# Halogenation Activity of Mammalian Heme Peroxidases

**DOI:** 10.3390/antiox11050890

**Published:** 2022-04-30

**Authors:** Jürgen Arnhold, Ernst Malle

**Affiliations:** 1Medical Faculty, Institute of Medical Physics and Biophysics, Leipzig University, 04107 Leipzig, Germany; 2Gottfried Schatz Research Center, Division of Molecular Biology and Biochemistry, Medical University of Graz, 8010 Graz, Austria

**Keywords:** cyanate, eosinophil peroxidase, hypobromous acid, hypochlorous acid, hypothiocyanite, lactoperoxidase, myeloperoxidase, peroxidasin, thyroid peroxidase

## Abstract

Mammalian heme peroxidases are fascinating due to their unique peculiarity of oxidizing (pseudo)halides under physiologically relevant conditions. These proteins are able either to incorporate oxidized halides into substrates adjacent to the active site or to generate different oxidized (pseudo)halogenated species, which can take part in multiple (pseudo)halogenation and oxidation reactions with cell and tissue constituents. The present article reviews basic biochemical and redox mechanisms of (pseudo)halogenation activity as well as the physiological role of heme peroxidases. Thyroid peroxidase and peroxidasin are key enzymes for thyroid hormone synthesis and the formation of functional cross-links in collagen IV during basement membrane formation. Special attention is directed to the properties, enzymatic mechanisms, and resulting (pseudo)halogenated products of the immunologically relevant proteins such as myeloperoxidase, eosinophil peroxidase, and lactoperoxidase. The potential role of the (pseudo)halogenated products (hypochlorous acid, hypobromous acid, hypothiocyanite, and cyanate) of these three heme peroxidases is further discussed.

## 1. Introduction

In mammals, the phylogenetic peroxidase-cyclooxygenase superfamily of heme peroxidases currently comprises six members: myeloperoxidase (MPO), eosinophil peroxidase (EPO), lactoperoxidase (LPO), thyroid peroxidase (TPO), peroxidasin (PXDN), and peroxidasin-like protein (PXDNL) [1,2]. Common properties of these peroxidases are the presence of a heme group at the active site and the ability to oxidize (pseudo)halides under physiologically relevant conditions.

Among mammalian heme peroxidases, predominant attention is directed to MPO, EPO, and LPO, proteins involved in different aspects of immune reactions and inflammation. MPO and EPO are key components in the innate immune cell response of polymorphonuclear leukocytes (commonly termed neutrophils) and eosinophils, respectively. These cells are recruited to and activated at inflammatory loci, where both peroxidases participate in the inactivation and killing of pathogens [3,4]. LPO is secreted from epithelial cells at mucous surfaces and secretory glands, where it helps to maintain microorganisms in mucous linings and secretions at a low level [5]. In addition to their beneficial functions in immune defense, these peroxidases contribute via their products to the initiation and/or progression of disease, when proper controlling mechanisms towards cytotoxic agents are limited or exhausted [6,7].

TPO is the key enzyme for the production of iodine-containing hormones in the thyroid glands [8]. PXDN and PXDNL are only currently described. Whereas PXDN catalyzes the bromine-dependent formation of cross-links during the synthesis of collagen IV in connective tissues [9], the physiological role of PXDNL remains unknown [10].

In a series of reports dealing with heme peroxidases, the halogenation activity of these enzymes is solely related to the formation of hypohalous acids and hypothiocyanite (^−^OSCN). However, their activity is much broader and also includes substrate halogenation, different substrate modifications, and the formation of other halogenated species. In this review, we provide an overview about the halogenation activity of heme peroxidases, focus on physiologically relevant conditions of peroxidase-mediated halogenation reactions, and specify their contribution to health protection and the initiation and progression of disease.

## 2. Mechanisms of Halogenation Activity of Heme Peroxidases

### 2.1. The Heme Moiety of Heme Peroxidases

Heme b, also known as ferric protoporphyrin IX, is the key prosthetic group of mammalian heme peroxidases. To date, the molecular structures of these peroxidases are known from X-ray data with good resolution for MPO and LPO [11,12,13]. Whereas EPO, LPO, and PXDN are monomeric enzymes with one heme group, MPO and TPO form homodimers. In MPO, two identical subunits (each with one heme) are linked by a disulfide bridge. In contrast to other heme proteins such as hemoglobin or cytochromes, the heme is covalently linked to the apoprotein by two ester bonds in MPO, EPO, and LPO and, in the case of MPO, additionally by a sulfonium ion linkage [11,12,14]. The resulting heme curvature (which is the strongest in MPO) determines the extraordinary biochemical reactivity and redox properties of these peroxidases. As the peroxidase domain in mammalian heme peroxidases is highly conserved and also contains, in TPO and PXDN, the corresponding amino acid residues for ester linkages to the heme, it has been thought that, in TPO and PXDN, the heme is covalently coupled to the apoprotein moiety, too [15,16].

### 2.2. Activation and Major Catalytic Cycles of Heme Peroxidases

In the resting state of heme peroxidases, the heme iron is in the ferric state. To fulfill any halogenation activity, the ferric heme group (Por-Fe^3+^) has to be oxidized to Compound I (^+•^Por-Fe^4+^ = O), a state having two more oxidative equivalents than the resting enzyme. This is usually achieved by the reaction of the ferric enzyme with hydrogen peroxide (H_2_O_2_).
Por-Fe^3+^ + H_2_O_2_ ⟶ ^+•^Por-Fe^4+^ = O + H_2_O(1)

In this two-electron redox reaction, H_2_O_2_ is reduced to H_2_O, and the ferric heme is oxidized to Compound I [17], which is characterized by an oxo-ferryl moiety and the additional presence of a porphyryl cation radical [18]. The highly reactive Compound I is involved in two-electron oxidations of (pseudo)halides and also in one-electron oxidation of numerous substrates. In the latter reaction, Compound II is formed, where the heme bears an oxo-ferryl moiety, but no radical functions neither in the porphyrin ring nor in adjacent amino acid residues [17]. Thus, Compound II is in between Compound I and the resting enzyme concerning its redox state.

A spontaneous isoelectronic conversion of Compound I into Compound I* is known for LPO and TPO in the absence of suitable substrates [8]. In Compound I*, the radical moiety is not located on the porphyrin ring but on an adjacent amino acid residue. This heme form is, like Compound II, unable to oxidize (pseudohalides). 

In the halogenation cycle of heme peroxidases, the formation of Compound I (Equation (1)) is followed by Compound I-mediated oxidation of halides or thiocyanate (SCN^−^) under direct recovery of the resting enzyme (see Section 2.3). Compound I can also be converted into the resting enzyme via Compound II. The sequence resting enzyme → Compound I → Compound II → resting enzyme is known as the peroxidase cycle. In this cycle, both Compound I and Compound II oxidize suitable substrates by abstracting one electron. Redox conversion of purified resting peroxidase into Compound I can be followed by time-resolved UV–Vis spectroscopy. Upon this conversion, there is no shift in the wavelength maximum of the Soret band of the heme, but a significant decrease in absorption during the course of reaction. 

### 2.3. Reaction of Compound I with (pseudo)halides

For isolated MPO, EPO, and LPO, the conversion of Compound I into the resting state during the reaction of Compound I with (pseudo)halides can also be monitored by time-resolved characteristic absorbance changes coupled with sequential stopped-flow mixing. There is an increase in absorbance of the Soret band upon the addition of (pseudo)halides to preformed Compound I. By variation in (pseudo)halide concentrations, the application of conditions for first-order reactions, and verification that mono-exponential changes in absorbance values take place, second-order rate constants can be calculated for these redox reactions. Generally, for MPO Compound I, the highest second-order rate constant was found for the reaction with SCN^−^ followed by iodide (I^−^), bromide (Br^−^), and chloride (Cl^−^) at pH 7. At more acidic pH, rate constants are significantly higher [19]. EPO Compound I oxidizes Br^−^, I^−^, and SCN^−^ at neutral and slightly acidic pH values more powerfully than MPO. However, unlike MPO, Cl^−^ oxidation by EPO is, by far, less efficient [20]. LPO Compound I oxidizes I^−^ and SCN^−^ with a very high rate, but Br^−^ with a much lower rate [21].

Considering reaction rate constants and (pseudo)halide concentrations in the blood (0.11 M Cl^−^ [22], 40–110 µM Br^−^ [23], <0.1 µM I^−^ [24], 20–120 µM SCN^−^ [25,26]), it is reasonable to conclude that MPO primarily oxidizes Cl^−^, Br^−^, and SCN^−^ at neutral pH values. At 0.1 M Cl^−^, 100 µM SCN^−^, and pH 7, both ions are oxidized by MPO by nearly the same amount [27]. In albumin, tyrosine residues are preferentially brominated by the MPO-H_2_O_2_-halide system at physiological concentrations of Cl^−^ and Br^−^ and pH > 7 [28,29]. Br^−^ and SCN^−^ are the preferred (pseudo)halides oxidized by EPO. 

In epithelial cells of mucous surfaces and secretory glands, SCN^−^ and I^−^ are abundantly present as both ions are taken up from the circulation by an active transport mechanism via the sodium/iodide symporter [30]. Thus, in lining fluids and secretions, SCN^−^ is the prevailing species at neutral pH, being oxidized by LPO, and under inflammatory conditions even by MPO and EPO. SCN^−^ concentrations are around 0.5–4 mM in saliva [31,32], 300–450 µM in the nasal airway lining [33], and 270–650 µM in lung airway fluids [34].

The second-order rate constants reported thus far may be considered valuable data for a better understanding of (pseudo)halide oxidation. However, these data were obtained under non-physiological conditions as H_2_O_2_, and the corresponding (pseudo)halides were sequentially added to the respective peroxidase. In cells and tissues, halide ions and SCN^−^ are in equilibrium with peroxidases. The reversible binding of (pseudo)halide ions to both the resting enzyme and Compound I can affect (pseudo)halide oxidation by MPO and LPO [35,36,37]. A reverse-ordered sequential mechanism was proposed for LPO-mediated SCN^−^ oxidation to explain non-exponential kinetic traces at low SCN^−^ concentrations [38]. 

### 2.4. The Intermediary Halide–Compound I Complex

The detailed mechanism of redox conversion of Compound I to the ferric enzyme form is only scarcely known. Ferric MPO was only partially reconstituted from Compound I by the addition of Cl^−^ and Br^−^ at pH 7 in contrast to pH 5, where a total recovery of native MPO was found [19,39]. Apparently, an intermediary complex between Compound I and a halide was formed. Kinetic studies revealed that Cl^−^ forms a reversible high-spin complex with MPO Compound I [40,41]. This complex is also involved in the MPO-mediated formation of taurine chloramine [40]. Moreover, kinetic studies revealed that taurine is chlorinated by MPO without the formation of free hypochlorous acid (HOCl)/hypochlorite (^−^OCl). An enzyme-bound HOCl molecule was originally assumed to be the active agent during this chlorination process [40,42]. In the LPO-mediated iodination of tyrosine, an intermediary complex was reported to be formed between LPO Compound I and I^−^ [43]. Thus, considering the formation of an intermediary, reversible halide complex with Compound I, the actual redox process in the reaction of Compound I with a halide consists either in the decomposition of the halide–Compound I complex to the ferric enzyme and hypohalite, or in a reaction of the halide–Compound I complex with a small substrate that becomes halogenated [44] (Figure 1). 

Both mechanisms were evaluated in the MPO-mediated chlorination of small and more bulky substrates by careful kinetic examination of the resulting chloramine formation [44,45]. Investigation of pH effects revealed that the chlorinating species produced inside the heme cavity of MPO must be unprotonated [44]. According to the first mechanism, ^−^OCl is formed, which yields the chlorinating species HOCl after diffusion to the enzyme environment. Small substrates such as taurine can directly be chlorinated via the chloride–Compound I complex within the heme cavity. More bulky substrates such as the tripeptide Pro-Gly-Gly are chlorinated only outside the heme pocket via HOCl derived from ^−^OCl [44,45]. To what extent similar mechanisms are valid for other halides and other heme peroxidases still remains unknown. 

### 2.5. The Nernst Equation

The Nernst equation (Equation (2)) allows determining the reduction potential *E*’ of a redox couple, that is, the ability of the oxidant form of a redox couple to abstract one or more electrons from a substrate [46].
*E*’ = *E*’° + (*RT*/*nF*) ln(*a*_ox_/*a*_red_)(2)

In this equation, *E*’° represents the standard reduction potential, which refers to 1 M of all reactants of a pressure of 101.3 kPa in the case of gases. In life sciences, standard values are usually given at pH 7. These values are referenced to the potential of the standard hydrogen electrode, which is −0.42 V at pH 7. The gas constant *R* and the Faraday constant *F* are 8.31 J K^−1^ mol^−1^ and 96,485 As mol^−1^, respectively. The temperature is usually set to 298 K. The factor *n* is equal to the number of electrons transferred in a single reaction step between both partners of the redox couple. The values *a*_ox_ and *a*_red_ correspond to the activity products of all components involved in oxidation and reduction, respectively.

Generally, a redox reduction can thermodynamically proceed when the reduction potential of the reduction process is higher than the corresponding value for the oxidation reaction. Additionally, the rate of a redox reaction also depends on other factors such as steric hindrance, the availability of redox partners, and the stability of hydrate shells. 

### 2.6. Redox Properties of Conversions between Compound I and (pseudo)halides

The halogenation cycle of heme peroxidases consists of two redox reactions. By means of Equation (1), Compound I is formed. In the second reaction, Compound I is reduced to the ferric enzyme form, and the (pseudo)halide is oxidized to the corresponding hypo(pseudo)halous acid or by incorporation into an adjacent small substrate (Figure 1). In the description of redox properties, we focus here on the formation of hypohalous acids and hypothiocyanite (^−^OSCN).

During the halogenation cycle of heme peroxidases, the ferric enzyme alternates with Compound I. Written as a reduction process, the following half reaction results for this conversion:^+•^Por-Fe^4+^ = O + 2 e^−^ + 2 H^+^ ⟶ Por-Fe^3+^ + H_2_O(3)

The standard reduction potential of the redox couple Compound I/ferric MPO was determined to be 1.16 V at pH 7 [47]. The corresponding potentials for EPO and LPO are 1.10 V [47] and 1.09 V [48], respectively. As two electrons and two protons are involved in this half reaction, these potentials increase according to Equation (2) by 0.06 V per unit decreasing pH. 

Compound I of MPO, EPO, and LPO is able to oxidize (pseudo)halides to hypohalous acids and ^−^OSCN. At pH 7, the following half reactions result for these oxidations (written as a reduction process):HOCl + 2 e^−^ + H^+^ ⟶ Cl^−^ + H_2_O,(4)
HOBr + 2 e^−^ + H^+^ ⟶ Br^−^ + H_2_O,(5)
HOI + 2 e^−^ + H^+^ ⟶ I^−^ + H_2_O, and(6)
^−^OSCN + 2 e^−^ + 2 H^+^ ⟶ SCN^−^ + H_2_O.(7)

In these half reactions, HOCl, HOBr, and HOI are given in their protonated form, as their p*K*_a_ values are 7.53 [49], 8.8 [50], and 10.0 [51], respectively. For hypothiocyanous acid (HOSCN), two p*K*_a_ values were reported, namely, 5.3 [52] and 4.85 [53]. The standard reduction potential for the redox couple HOCl/Cl^−^, H_2_O is 1.28 V at pH 7 [54]. The corresponding standard values for the redox couples HOBr/Br^−^, H_2_O and HOI/I^−^, H_2_O are 1.13 V and 0.78 V, respectively [54]. The lowest value with 0.56 V was determined for ^−^OSCN/SCN^−^, H_2_O [54]. Below pH 7, the potentials for Cl^−^, Br^−^, and I^−^ oxidation increase according to Equation (2) by 0.03 V per unit decreasing pH, as two electrons and one proton are involved in these half reactions.

In I^−^ and SCN^−^ oxidation by heme peroxidases, which proceeds with a high rate, there is a great difference between the reduction potentials of the involved redox couples. Another situation exists in Cl^−^ and Br^−^ oxidation by MPO and EPO [47,55]. Both potentials for reduction and oxidation differ only slightly even when actual reactant concentrations are considered. Moreover, as shown before, the reduction potentials for the couple Compound I/ferric MPO or EPO exhibit another pH dependence as reported for the redox couples HOCl/Cl^−^, H_2_O and accordingly HOBr/Br^−^, H_2_O. Consequently, the formation of these hypohalous acids by MPO or EPO is only possible below a certain pH threshold that is dependent on the respective halide concentration [54,55]. 

At 0.1 M Cl^−^ and pH 7, the MPO-H_2_O_2_-Cl^−^ system is unable to induce chlorohydrin formation in unsaturated phosphatidylcholines [55,56] or to cause an accumulation of diene conjugates in low-density lipoprotein (LDL) particles [57]. A certain pH threshold value exists for these reactions at lower pH. At 0.1 M Cl^−^, these pH thresholds were 6.5 for MPO and 6.0 for EPO during the formation of chlorohydrins [55]. Otherwise, taurine is applied up to pH 8 to detect the chlorination activity of MPO [36,58,59,60]. There is no discrepancy in these data regarding the existence of a pH threshold for HOCl formation by the MPO-H_2_O_2_-Cl^−^ system, as taurine chlorination by MPO can occur without the participation of free HOCl [44]. Concerning the redox process of the MPO-mediated taurine chloramine formation, the reduction of MPO Compound I (Equation (3)) is directly linked with the redox couple taurine chloramine/taurine, Cl^−^. The standard reduction potential of this couple is unknown. It should be lower than the potential for HOCl/Cl^−^, H_2_O, as the reagent HOCl is well known to oxidize taurine, a reaction commonly used to estimate the chlorination capacity of HOCl added as a reagent or generated by the MPO-H_2_O_2_-Cl^−^ system.

Maybe a similar assumption is valid for monochlorodimedon, which is also applied to follow the chlorinating activity of MPO at neutral pH. The chlorination of this agent by chloroperoxidase is assumed to occur via an intermediary complex formation via so-called enzyme-bound HOCl [61]. The detailed mechanism of interaction between monochlorodimedon and MPO Compound I remains unknown. 

## 3. Products of the Halogenating Activity of Heme Peroxidases

Talking about the products of the halogenating activity of heme peroxidases, a distinction should be made between (i) the halogenation of selected substrates adjacent to heme peroxidases and (ii) the formation of free (pseudo)halogenated species, in particular hypohalous acids and ^−^OSCN (Figure 2). 

### 3.1. Halogenation of Selected Substrates

Within these reactions, halide ions are oxidized by activated peroxidases, and the oxidized halide (having the formal oxidation state +1) is directly transferred to the corresponding substrate that is bound to the enzyme. Transfer reactions are known for I^−^, Br^−^, and Cl^−^. The resulting halogenated substrates can fulfil important physiological functions as hormones or play a role as intermediate products during basement membrane formation. In some transfer reactions, peroxidase-bound hypohalites are assumed as intermediary species.

In thyroid follicular cells, I^−^ is enriched by an active transport mechanism via the sodium/iodide symporter [62]. TPO, which is located in the follicular lumen and anchored at the apical membrane of follicular cells, uses H_2_O_2_ and I^−^ to iodinize tyrosine residues in thyreoglobulin. In a second step, TPO catalyzes the phenolic coupling of two iodinized tyrosine residues [16]. After proteolysis of the iodinized tyrosine and dimerized tyrosine residues from thyreoglobulin, different iodine-containing products including the thyroid hormones triiodothyronine (also known as T3) and thyroxine (T4, previously called tetraiodothyronine) are formed. During this iodination step, hypoiodite bound to TPO Compound I is thought to be the active species [8]. 

PXDN, originally identified in Drosophila, may be considered an enzyme-matrix protein. Besides the peroxidase domain, which is homologous with human MPO and EPO, several extracellular matrix motifs are present in the primary structure of PXDN [63]. This allows PXDN to bind to collagen IV protomers and to induce, by means of Br^−^ oxidation, sulfilimine cross-links between methionine and hydroxylysine residues during the synthesis of basement membranes [64,65]. In the first step, a methionine residue is converted into a bromosulfonium intermediate that favors a cross-link to the amino group of an adjacent hydroxylysine residue. Oxidation of Cl^−^ by PXDN instead of Br^−^ is, by far, less effective in the formation of collagen IV cross-links [64]. In these reactions, both HOBr and HOCl are assumed to be reactive species [63,64]. The potential role of HOBr as an active agent is further supported by the fact that PXDN mediates, in addition to collagen cross-links, the formation of 3-bromotyrosine in proteins in cells expressing PXDN [66].

Another example is the MPO-mediated chlorination of taurine (see Section 2.4 and Section 2.6). Careful investigation of the fine mechanism of this reaction revealed that free HOCl does not participate within this redox conversion [44]. The physiological relevance of taurine chlorination by MPO remains puzzling. In resting neutrophils and also during the early steps of neutrophil-derived phagocytosis, MPO and taurine are well separated from each other. MPO is present in azurophilic granules and the phagosomal compartment, while taurine is located in the cytoplasm at a concentration between 22 and 26 mM [67]. In undergoing neutrophils, both MPO and taurine can exist in the same compartment. Under these conditions, MPO is involved as an essential component in the formation of DNA-based extracellular traps [68]. It remains unclear whether taurine chloramine plays an active role in neutrophil extracellular trap (NET) formation or whether its formation protects other cell components from the toxic effects of HOCl.

Maybe some other substrates will also be halogenated without the formation of free hypohalous acids after binding near the active site of heme peroxidases.

### 3.2. Formation of Free (pseudo)halogenated Species

#### 3.2.1. Thiocyanate Oxidation Products

SCN^−^ is easily oxidized by Compound I of MPO, EPO, and LPO. The major oxidation product ^−^OSCN is in equilibrium with its protonated form HOSCN, having a p*K*_a_ value of 5.3 [52] or 4.85 [53]. 

At inflammatory sites, an alternative pathway for the formation of ^−^OSCN/HOSCN consists in the reaction of HOCl or HOBr with SCN^−^. As SCN^−^ is very rapidly oxidized by HOCl [69] and HOBr [70], SCN^−^ can efficiently compete with other substrates for these powerful hypohalous acids. As addressed in several reports, SCN^−^ is able to protect cell and tissue components from the devastating actions of MPO-generated HOCl [71,72,73,74].

A more hydrophobic local environment at the active site of heme peroxidases might apparently favor the formation of thiocyanogen ((SCN)_2_) and thiocarbamate-(*S*)-oxide, a decomposition product from (SCN)_2_ [75]. Alternatively, ^−^OSCN/HOSCN is known to decay by dismutation and oxidation reactions to sulfate, hydrocyanic acid, and cyanate [76,77,78]. In these reactions, cyanosulfurous acid (HO_2_SCN) and cyanosulfuric acid (HO_3_SCN) are transient intermediates. 

Among the SCN^−^ oxidation products, major focus is directed to reactions of ^−^OSCN/HOSCN and cyanate with the respective substrates under (patho)physiological conditions. 

#### 3.2.2. Iodide Oxidation Products

Besides TPO, I^−^ can also be oxidized by MPO, EPO, and LPO. However, due to the low abundance/concentration of I^−^ in the circulation (<0.1 µM [24]) and other body fluids, oxidation of I^−^ by these three peroxidases does not play a significant role in the human organism. However, in biotechnological applications, I^−^ is often utilized as a substrate for LPO [79,80]. 

During I^−^ oxidation, a variety of products are formed including molecular iodine (I_2_), triiodide anion (I_3_^−^), HOI, and ^−^OI [81]. HOI dominates as the main oxidation product at micromolar concentrations only in the pH range from 8.4 to 9.3. Thus, in microbial killing, the major iodide oxidation products are I_2_ and I_3_^−^ at neutral pH values [81].

#### 3.2.3. Chloride Oxidation Products

At neutral and slightly acidic pH values, only MPO oxidizes Cl^−^ at reasonable rates. However, the formation of free HOCl via MPO is restricted to acidic pH values below 6.0–6.5. This pH threshold results from the redox properties of the involved redox couples during the formation of HOCl (see Section 2.6).

The primary species formed in the heme cavity during MPO-mediated Cl^−^ oxidation is obviously ^−^OCl [44]. After its diffusion from the heme cavity to the enzyme environment (see Figure 1), protonation of ^−^OCl leads to the formation of HOCl and subsequently molecular chlorine (Cl_2_ (aq)) according to the following equilibria: ^−^OCl + H^+^ ⇌ HOCl and(8)
HOCl + H^+^ + Cl^−^ ⇌ Cl_2_(aq) + H_2_O.(9)

The p*K*_a_ value of HOCl is 7.53 [49]. For Equation (9), an equilibrium constant of 1.04 × 10^−3^ M^2^ at 25 °C and an ionic strength of 0.5 M was reported [82]. Thus, the formation of substantial Cl_2_ becomes evident only at strong acidic pH values.

Both HOCl and Cl_2_ are powerful species to promote substrate chlorination and oxidation. In one of the first reports about Cl^−^ oxidation via MPO, the authors did not differentiate between HOCl and Cl_2_ as suitable chlorinating species [83]. In some reports, the ability of MPO to produce Cl_2_ was demonstrated via the Cl_2_-mediated formation of 3-chlorotyrosine [84], 5-chloro-2′-deoxycytidine from 2′-deoxycytidine [85], 5-chlorocytosine in bacterial RNA [85], or even chlorinated sterols [86]. 

Interestingly, the MPO-H_2_O_2_-Cl^−^ system oxidizes L-tyrosine to two different products as a consequence of the pH-dependent formation of Cl_2_ and/or HOCl [84]. Below pH 5, Cl_2_-mediated formation of 3-chlorotyrosine dominates, whereas the HOCl-mediated pathway leads to the formation of *p*-hydroxyphenylacetaldehyde via an intermediary monochloramine. The latter product can be formed within a pH range from 4 to 8.

#### 3.2.4. Bromide Oxidation Products

Both MPO and EPO are known to oxidize Br^−^. However, it remains unclear whether a similar mechanism to that proposed for Cl^−^ oxidation exists for the oxidation of Br^−^ (see [44]). HOBr is generally believed to be the major Br^−^ oxidation product generated by MPO and EPO. The following equilibria can be formulated:^−^OBr + H^+^ ⇌ HOBr and(10)
HOBr + H^+^ + Br^−^ ⇌ Br_2_(aq) + H_2_O.(11)

The p*K*_a_ value of HOBr is 8.8 [50]. The equilibrium constant for molecular bromine (Br_2_(aq)) hydrolysis equals 6.1 × 10^−9^ M^2^ at 25 °C and an ionic strength of 0.5 M [87]. Thus, HOBr is the dominating species in neutral and slightly acidic aqueous media. Considering a Br^−^ concentration of 10^−4^ M, around and below pH 5, the formation and contribution of Br_2_ are obvious.

#### 3.2.5. Oxidation of Selenocyanate

Selenocyanate (SeCN^−^) functions as an intrinsic selenium pool in different mammalian cells [88,89]. This anion can apparently accumulate in mucous lining fluids and secretions by the same active transport mechanism reported for SCN^−^ [90]. The LPO-H_2_O_2_ system is able to oxidize SeCN^−^, but less efficiently than published for SCN^−^ [91]. However, the resulting ^−^SeOCN is more potent in the killing of microbes (such as *Pseudomonas aeruginosa*, *Burkholderia cepacia complex*, and methicillin-resistant *Staphylococcus aureus*) when compared to the ^−^OSCN/HOSCN system [91]. 

#### 3.2.6. Formation of inter(pseudo)halogens

Activated heme peroxidases are principally able to produce inter(pseudo)halogens, where two different (pseudo)halogens are coupled within one molecule. However, their pathophysiological relevance is rather limited and widely unknown. The brominating agent bromine chloride (BrCl) was postulated to be generated by the MPO-H_2_O_2_-Cl^−^/Br^−^ system [92]. The formation of cyanogen iodide (ICN) by MPO or LPO occurs when I^−^ is present in excess over SCN^−^ [93,94], a condition that is far from that observed in biological fluids. Finally, cyanogen chloride (ClCN) and cyanogen bromide (BrCN) may be formed by the reaction of HOCl or HOBr with cyanide (CN^−^) [95,96].

## 4. (Patho)Physiological Relevance of Reactions of Hypohalous Acids and Thiocyanate Oxidation Products

### 4.1. Heme Peroxidases in Immune Reactions

The heme peroxidases MPO, EPO, and LPO are part of the immune defense system in living organisms. Several major functions of these heme peroxidases are discussed [6,7]. In newly formed phagosomes of neutrophils recruited to inflammatory sites, MPO is apparently involved in the rapid pH increase, thus creating optimal conditions for the destructive action of either serine proteases or other microbicidal agents [6,97]. Dying neutrophils are known to release so-called NETs, where MPO as an essential element and other cationic proteins are tightly associated with DNA [68,98,99]. These NETs inactivate hyphenated fungi and microbes independent of phagocytosis [100,101]. In addition to these beneficial functions, the highly cationic charged MPO protein is known to form complexes with several acidic proteins and polymers after its release from neutrophils [28,102,103,104,105,106,107,108]. This can largely affect physiological functions. For example, MPO transcytoses through endothelial cells of blood vessels, associates closely with fibronectin at the basolateral side, diminishes thus the bioavailability of nitric oxide (NO) at this location, and impairs the NO-mediated vessel relaxation [109,110]. Further, the attachment of MPO to cell surface epitopes of the inflamed endothelium can induce the formation of antibodies against MPO [111]. These MPO-antineutrophil cytoplasmic antibodies are involved in the pathogenesis of glomerulonephritis and vasculitis of the upper and lower airways [112,113].

At sites of inflammation, eosinophils are recruited and activated together with other cells such as mast cells and basophils involved in reactions of the type 2 immune response [114]. Larger pathogens such as helminths and other parasites as well as virus-infected and cancer cells are targeted by eosinophils [115,116,117]. In targeted cells, eosinophil granule proteins including EPO exhibit cytotoxic activities [4]. Eosinophils are involved in different allergic diseases, where EPO contributes with substrate bromination and carbamylation to disease progression [118,119,120,121]. Eosinophils are also known to release DNA-containing extracellular traps [122,123] and, in contrast to neutrophils, free extracellular granules that can target conidia from *Aspergillus fumigatus*, a common fungus in allergic bronchopulmonary mycoses [124,125].

In mucous fluids and secretions, the main function of the LPO-H_2_O_2_-SCN^−^ system is the generation of the microbicidal ^−^OSCN/HOSCN to maintain pathogen contamination of a low level [5,7].

### 4.2. Control over Tissue Damage by Heme Peroxidases

Immune defense reactions are usually directed to combat against unwanted and/or unpredictable invaders (such as microbes, fungi, and parasites), to eliminate damaged cell and tissue material, and to recognize and kill virus-laden and tumor cells. In inactivation of pathogens, immune cells, especially neutrophils and eosinophils, use numerous aggressive components that can also principally damage intact host cells. In healthy organisms, the release of cytotoxic elements from immune cells or damaged host cells is antagonized by protective principles that resist and inactivate these destructive agents [6,126]. This balance between cytotoxic invaders and protective defender activities can be disturbed under very strong acute and long-lasting chronic inflammatory conditions due to a decreased capacity or exhaustion of protective mechanisms. The latter can vary in a wide range from one patient to another. Thus, the limited ability of host antagonizing principles favors chronic inflammatory states and provides the basis for disease progression.

Among the antagonizing principles (examples are given in [6]), the plasma protein ceruloplasmin forms a high-affinity inhibitory complex with MPO, and also with EPO, by insertion of a polypeptide loop into the heme pocket of these heme peroxidases [127,128,129,130]. SCN^−^ is known to inactivate HOCl and HOBr [69,70]. In addition, a defense against HOCl and HOBr is given by reduced glutathione (GSH), ascorbate, and urate [131]. Glutathione also inactivates an excess of ^−^OSCN/HOSCN [132]. Further, the availability of H_2_O_2_ can be diminished by several H_2_O_2_-consuming proteins such as peroxiredoxins, catalase, and glutathione peroxidase [133,134,135]. An overview about antagonizing principles against components and major products of the heme peroxidase-H_2_O_2_-(pseudo)halide system is presented in Figure 3. To sum up, insufficient antagonizing mechanisms favor substrate halogenation and oxidation by heme peroxidases. 

### 4.3. Important Reactions of (pseudo)hypohalous Acids and Targets of MPO and EPO

Reactions of HOCl [136,137], HOBr [138], and ^−^OSCN/HOSCN [139,140] with different biologically relevant substrates have been intensively investigated at neutral pH values. Whereas HOCl and HOBr prefer a broad range of substrates, ^−^OSCN/HOSCN reacts more specifically with substrates containing accessible sulfhydryl and selenocysteine residues [131]. Cysteine and methionine residues of proteins and GSH are preferred targets for HOCl [141]. Concerning protein residues, HOBr reacts well with cysteine, methionine, and tryptophan residues, and also, with a sufficiently high rate, with cystine, lysine, tyrosine, histidine residues, and α-amino groups [131]. Interestingly, the reaction rate of tyrosine residues with HOCl is about five orders of magnitude lower than that with HOBr [138].

The aforementioned data were obtained on isolated, artificial systems. After their release from invading leukocytes, heme peroxidases are often attached to proteins, cell surfaces, lipoproteins, and components of the extracellular matrix. This attachment favors local reactions of MPO and EPO products with lipids, nucleic acids, and carbohydrates. In undergoing neutrophils, MPO resides on the cell surface at phosphatidylserine epitopes [142]. Comparable high rates were reported for the reaction of HOCl and especially of HOBr with the serine and ethanolamine groups of phospholipids [131]. Plasmalogens, a class of ether phospholipids abundantly present in the heart and brain and known to trigger either an anti- or proinflammatory response [143], are rapidly targeted by HOCl and HOBr on the double bond adjacent to the ether moiety [144,145,146], where cleavage of the vinyl ether bond at the *sn*-1 position results in the formation of a lysophospholipid and an α-halogenated fatty aldehyde [147]. 

MPO binds to lipoproteins such as LDL [148,149,150] (where apoB-100, the major apolipoprotein of LDL, and respective lipid species represent major targets for HOCl attack [151,152]) and high-density lipoprotein (HDL) [153,154,155] (quite heterogenous lipoprotein particles varying in density, size, electrophoretic mobility, and protein and lipid composition; for a review, see [156]). Although electron microscopy studies originally revealed limited binding and internalization of MPO (compared to EPO) to inflammatory cells [157], ongoing studies clearly underscore the adverse oxidative reactions of endothelial-localized MPO [158] as well as endothelial-transcytosed MPO [109] leading to targeted subendothelial matrix oxidation of proteins [159,160,161,162,163,164] via MPO-mediated nitration and/or chlorination; the latter reaction could by suppressed by SCN^−^ and nitrite (NO_2_^−^) [162]. 

### 4.4. Reactions of ^−^OSCN/HOSCN

In unperturbed mucous lining fluids and secretions, the LPO-H_2_O_2_-SCN^−^ system contributes to the control over microorganisms by producing ^−^OSCN/HOSCN. The uncharged HOSCN can permeate through biological membranes and thus target intracellular GSH as well as critical thiol and selenocysteine residues of proteins [165,166]. It penetrates even into biofilms [165,166]. In epithelial cells of mucous surfaces, HOSCN is inactivated by thioredoxin reductases [167]. Bacteria are unable to inactivate HOSCN in this way [167].

Both in asthma patients and in a murine asthma model, overproduction of ^−^OSCN/HOSCN by peroxidases can promote allergic inflammation in the lung airways [168,169,170]. In airway epithelial cells, HOSCN activates the transcription factor NFκB via protein kinase A, induces necrotic processes, and favors the release of IL-33 and other proinflammatory mediators [170,171,172,173]. Accordingly, eosinophils are recruited to and activated at inflammatory sites [174]. ^−^OSCN/HOSCN, IL-13, and IL-33 are assumed to be components of a vicious circle that exaggerates and prolongs the type 2 immune response in allergic diseases [171]. 

To sum up, adverse reactions of excess ^−^OSCN/HOSCN are favored by enhanced SCN^−^ levels in the blood, recruitment of eosinophils to inflammatory sites, release of EPO from these cells, and a diminished glutathione level in mucous fluids.

### 4.5. Reactions of Cyanate

^−^OCN was identified as a minor product of SCN^−^ oxidation by MPO and more efficiently by EPO [78]. There are two principal routes for the oxidation of SCN^−^, the redox conversion of SCN^−^ by Compound I of heme peroxidases, and the reaction of SCN^−^ with HOCl or HOBr. In both routes, the formation of SCN^−^ oxidation products including ^−^OCN is favored by high SCN^−^ levels. Furthermore, ^−^OCN can also be derived from urea, as both substances are in equilibrium with each other [175]. Thus, conditions favoring an increase in urea such as uremia also contribute to enhanced levels of ^−^OCN [176,177]. In other words, the resulting carbamylation of biological targets is not specific to heme peroxidase-mediated posttranslational modifications.

^−^OCN promotes the carbamylation of functional residues in proteins. Although different amino acid residues are targeted by ^−^OCN, lysine residues are preferred. In the latter case, this residue is converted into a homocitrulline moiety. Carbamylation affects several physiological functions. It favors T cell activation [178] and promotes endothelial dysfunction [179]. Carbamylated proteins were detected in atherosclerotic plaques adjacent to MPO [180], in dysfunctional HDL induced by MPO [181], and at inflammatory sites of eosinophil-driven allergic asthma [120]. The homocitrulline level in the blood correlates well with an increased risk of cardiovascular disease [180].

Additional pathways of MPO-mediated carbamylation reactions were recently described under the involvement of ^−^CN [182]. ^−^OCN is formed as a result of the oxidation of ^−^CN by MPO Compound I or by the reaction of HOCl with ^−^CN. Otherwise, carbamylation is also induced via the reaction of ^−^SCN with chloramines.

### 4.6. Bromination of Substrates

At inflammatory sites, the bromination of substrates is mostly attributed to the activity of EPO. As MPO is also able to produce reactive brominating species, the detection of these products cannot be used as a biomarker for the brominating activity of EPO. Nevertheless, the yield of brominated products increases under conditions of eosinophilia, when eosinophils are massively recruited to inflamed loci. An overview about the formation of brominated products in biological systems is presented in Table 1. 

In proteins of the airway epithelium of asthmatic patients, 3-bromotyrosine, 3,5-dibromotyrosine, and 3-nitrotyrosine were detected [122,183,189]. In the liver and urine of lipopolysaccharide (LPS)-treated rats, enhanced levels of 8-bromo-2′-deoxyguanosine and 3-bromotyrosine were reported [188]. Diabetic patients excreted increased values of 8-bromo-2′-deoxyguanosine in their urine [188].

### 4.7. Chlorination of Substrates

Among heme peroxidases, MPO is the only protein able to catalyze the generation of HOCl or Cl_2_ and to further chlorinate a variety of biological substrates. The incorporation of a chlorine atom into a substrate is of special interest, as this unequivocally demonstrates MPO-mediated chlorination capacity, which can be used as a biomarker for the chlorination activity of MPO. Indeed, there are numerous reports about the substrate chlorination of cholesterol, DNA, and pyrimidine nucleotides, and secondary reactions of chlorinated pyrimidines with GSH, NADH, tertiary amines, and a panel of other biological substrates [190,191,192,193,194], as well as substrate chlorination in cell and tissue components under pathophysiological conditions. Selected examples are presented in Table 2.

Only a few chlorinated products were detected in biological samples under disease conditions. The formation of 3-chlorotyrosine is regarded as a minor product of the MPO-H_2_O_2_-Cl^−^ system [198]. Nevertheless, enhanced levels of 3-chlorotyrosine were detected in atherosclerotic lesions [199], HDL associated with atherosclerotic plaques [200,201], and LDL [202].

Plasmalogen-derived α-chloro fatty aldehydes are elevated in human aortic atherosclerotic plaques [203] as well as in infarcted rat myocardium [204]. These α-chloro fatty aldehydes contribute to tissue damage by several mechanisms including endothelial dysfunction, inhibition of endothelial NO synthase activity, and the promotion of myocardium contractile and blood–brain barrier dysfunction [199,205,206,207,208]. Plasmalogen-derived 2-chlorohexadecanal is generated in the mouse heart during endotoxemia, and treatment of murine HL-1 cardiomyocytes leads to the conversion of this chloro fatty aldehyde to 2-chlorohexadecanoic acid and 2-chlorohexadecanol [209]. Bromine inhalation mimics ischemia/reperfusion cardiomyocyte injury in rats via the intermediate formation of 2-bromo fatty aldehyde (2-bromohexadecanal) [210] apparently via the action of MPO and EPO [211].

Several chlorinated products of nucleobases were detected after incubation with the MPO-H_2_O_2_-Cl^−^ system [85,185,191]. Of special interest is the confirmation of these products in inflamed tissue material. 5-Chlorouracil was found in atherosclerotic plaques [196]. Like 8-bromo-2′-deoxyguanosine, enhanced values for 8-chloro-2′-deoxyguanosine were detected in the liver and urine of LPS-treated rats as well as in the urine of diabetic patients [188].

### 4.8. Identification of Chlorinated Epitopes/Proteins in Biological Specimens

Basically, polymorphonuclear neutrophils are the major reservoir of MPO (the most abundant peroxidase in humans), representing 2–5% of cell weight, while the level of MPO in monocytes is approximately 1% [212,213], but MPO is also present in certain macrophage subtypes, including liver (Kupfer cells) and brain macrophages (microglia; [214]). While the use of different mass spectrometry analysis techniques (see Table 1 and Table 2) has turned out to be suitable for the identification of EPO/MPO-derived chlorinated and/or brominated species, immunological techniques may be considered crucial, focusing on MPO-generated chlorinated epitopes under inflammatory and/or disease conditions. 

Immunohistochemical staining of chlorinated proteins generated by the MPO-H_2_O_2_-Cl^−^ system has been performed with the use of specific monoclonal antibodies (mAbs, clone 2D10G9 [215], and clones 6E10A11 and 10A7H9 [202]) in various inflammatory conditions in humans such as kidney disease (glomerular and tubulointerstitial inflammatory and fibrotic lesions [216], membranous glomerulonephritis [214,217], and acute tubular damage [218]), placental tissues (first trimester following complications [219] and third trimester in normal pregnancy [220]), liver tissue (nonalcoholic steatohepatitis [221]), myocardial heart tissue (infarcted human left ventricle [222]), and in particular atherosclerotic lesion material (autopsy material, lesion types II to VI [105,107,202,223,224,225,226]). Most importantly, the mAbs raised against in vitro chlorinated proteins not only detected HOCl-modified proteins such as modified apoB-100 [223] and the modified matrix glycoprotein fibronectin [107] in situ but also detected these modified proteins when they were extracted from lesion material and subjected to immunoblot analyses: data that confirm enzymatic in vivo chlorination via the MPO-H_2_O_2_-Cl^−^ system. Extensive staining for HOCl-modified proteins (colocalizing with MPO and/or mononuclear cells, similar to that observed in the human system) was also found in atherosclerotic lesions from homozygous [227] and heterozygous Watanabe heritable hyperlipidemic rabbits [228], but to a lesser extent in lesions from a specific strain of New Zealand White rabbit with a high atherosclerotic response to hypercholesterolemia [227].

Although the expression of granule enzymes (including MPO) is much lower in the rodent system compared to humans [229,230], substantial staining for HOCl-modified proteins/epitopes has been detected in C57BL/6 murine livers following galactosamine/endotoxin-induced hepatoxicity: data in line with the increased staining for 3-chlorotyrosine adducts [231], another footprint for the generation of HOCl by neutrophil-derived MPO [232]. Pronounced staining for HOCl-modified epitopes became apparent during hepatic ischemia/reperfusion injury in C57BL/6 mice [233] to an extent similar to that observed during hepatic ischemia/reperfusion injury in male Sprague Dawley rats [234]. 

Indeed, progressive neutrophil accumulation and activation in the liver following treatment with monocrotaline, a pyrrolizidine alkaloid, have been paralleled by an increased activity of MPO and pronounced staining of HOCl-modified proteins/epitopes in Sprague Dawley rats [235]. Furthermore, different combinations of LPS with other drugs were applied to follow neutrophil-induced liver injury in Sprague Dawley rats. (i) Cotreatment with ranitidine, a H_2_ histamine receptor antagonist, increased staining of HOCl-epitopes (in the midzonal areas of liver lobules [236]) that could be significantly impaired by the mitogen-activated protein kinase inhibitor SB 239063 [237]. (ii) Cotreatment with sulindac (a nonsteroidal anti-inflammatory drug) elevated the panlobular formation of HOCl-modified epitopes [238] that could be inhibited by etanercept, a tumor necrosis factor inhibitor, or plasminogen activator inhibitor-1. (iii) Cotreatment with amiodarone, an antiarrhythmic medication, after a 10 h treatment led to massive accumulation of HOCl-modified epitopes [239] that could be reduced to baseline levels by either rabbit antineutrophil antiserum or heparin. Colocalization of chlorinated epitopes with lectin-like oxidized LDL receptor-1 during the initiation of transplant vascular injury in rats [240] is supported by data that the lectin-like oxidized LDL receptor-1 acts as a receptor for HOCl-modified lipoproteins [241,242]. Most importantly, high oral dosing with SCN^−^, a competitive MPO substrate, protected against myocardial ischemia/reperfusion injury in male Sprague Dawley rats by impairing infarct size and decreasing mAb recognition of HOCl-damaged myocardial proteins generated via the MPO-H_2_O_2_-Cl^−^ system of activated neutrophils [243]. Whether the addition of NO_2_^−^ may lead to an additional decrease remains to be elucidated. It is worth mentioning that NaSCN treatment has been found to attenuate atherosclerotic plaque and chlorotyrosine formation and to improve endothelial regeneration in apoE^−^/^−^ mice [72]. In another mouse model (LDL receptor^−^/^−^ mice) expressing human MPO, SCN^−^ supplementation has been reported to decrease atherosclerotic plaque formation and to improve endothelial vasodilatation [71].

### 4.9. Role of pH in Chlorination Activity of MPO at Inflammatory Sites

The presence of chlorinated species at inflammatory sites raises several questions about the conditions favoring these modifications. First, the detection of chlorinated material is indicative of a weak status of antagonizing protective mechanisms. As outlined in Section 4.2., several major mechanisms which control concentrations of H_2_O_2_ and inactivate heme peroxidases and their products are known. Second, after its release from undergoing neutrophils, the highly cationic MPO may bind to acidic epitopes of proteins, lipoproteins, cell surfaces, and extracellular matrix components, resulting in HOCl-mediated modifications of these targets. Third, a pH decrease below 6 favors the MPO-mediated formation of HOCl and Cl_2_ and thus chlorination reactions as a further consequence. Several reports exist demonstrating that the bulk pH might be lower at inflammatory sites in contrast to healthy tissue regions (as reviewed in [244]). The reasons for this deviation are metabolic acidosis, and diminished buffer capacity. For example, in cystic fibrosis patients, the pH value of inflamed airway mucous fluids was 0.3–0.5 pH units lower than that measured in unperturbed fluids [245,246,247]. 

As the pH value is usually measured by microelectrodes or pH-sensitive fluorophores in biopsy specimens, a mean pH value is given without consideration of local pH deviations. Protons and other cations can be enriched in the near environment of acidic artificial polymers and DNA [248,249]. Model calculations revealed a distance-dependent decrease in the local pH at the surface of DNA origami. Within a distance of 5 nm, the pH falls by about two pH units of the bulk value [249]. Intriguingly, horseradish peroxidase and glucose oxidase attached to a DNA scaffold exhibited enhanced activity as a result of this pH decrease [249]. Similar pH effects were reported for polyelectrolyte-bound trypsin and chymotrypsin [250,251]. After recruitment of neutrophils to inflammatory sites, complexes between MPO and DNA are very common from undergoing neutrophils [98,99]. At present, we can only speculate about the local pH conditions at the surface of MPO attached to DNA. 

A further aspect is important for the presence of acidic pH values. At inflammatory loci, undergoing neutrophils and tissue components are scavenged by macrophages. Unlike classically activated human macrophages (type 1 macrophages), alternatively activated human macrophages (type 2 macrophages) acidify their phagosomes to pH 5.0 within 10 min [252,253]. Macrophage phagosomal pH values around and below pH 5 were also reported by other groups [254,255]. As MPO is present in the material taken up by macrophages at inflammatory sites, good conditions are created for MPO-mediated chlorination reactions inside the phagosome. 

## 5. Conclusions

The formation of PXDN-mediated bromine-dependent cross-links in collagen IV and the iodination of tyrosine residues in thyreoglobulin by TPO represent functional reactions of specialized heme peroxidases. Their halogenation activities are embedded in complex processes of the synthesis of vascular basement membranes and endocrine hormones. Both heme enzymes incorporate oxidized halides into substrates tightly associated with these proteins. For MPO, EPO, and LPO, halogenation of bound substrates is evidenced only for the chlorination of taurine by MPO at present. In contrast to PXDN and TPO, in their halogenation activity, MPO, EPO, and LPO apply small (pseudo)halogenated species that can interact with multiple substrates. The halogenation activities of MPO, EPO, and LPO concern different aspects of immune defense reactions against pathogens as well as inactivation of virus-laden and other non-functional cells. Otherwise, these halogenation reactions can damage intact cell and tissue constituents and contribute to disease progression. These contrasting effects—on the one hand, playing an essential role in host defense and, on the other hand, contributing to the pathogenesis of long-lasting disorders—are well known for MPO [6] and also demonstrated for EPO [7]. The hypohalous acids, HOCl and HOBr, which are only generated by MPO and EPO, are the most reactive (pseudo)halogenated species. Thus, it is not surprising that, among mammalian heme peroxidases, both MPO and EPO are primarily involved in a panel of chronic inflammatory diseases. In addition, excess formation of SCN^−^ oxidation products also favors damaging reactions under pathological conditions.

Currently, we are far from a thorough understanding of detailed mechanisms of the (patho)physiological consequences of the halogenating activity of MPO and EPO. Of course, the activities of these peroxidases are closely associated with the recruitment and activation of neutrophils, eosinophils, and other inflammatory cells such as lymphocytes, monocytes, endothelial cells, fibroblasts, and platelets at local inflammatory sites. If we contemplate inflammatory processes as an indispensable defense mechanism against a wide range of different threats, health problems arise usually from very strong acute as well as long-lasting chronic inflammation. The latter is insufficiently terminated and concerns immunocompromised individuals most of all. Here, aggressive products released and generated from immune and undergoing host cells foment the inflammatory process in the long run. The heme peroxidases MPO and EPO and their products are part of this proinflammatory machinery when protecting mechanisms are limited or exhausted.

To summarize, the role of immunologically relevant heme peroxidases in defense reactions and disease development should be regarded in close context with other reactions and activities of immune cells. Revealing their biological activities under the inflammatory status in detail, it would be possible to understand the (patho)physiological relevance of the halogenation activity. It is still a great challenge for scientists to achieve a deeper understanding of these questions to extend therapeutic strategies based on basic molecular mechanisms.

## Figures and Tables

**Figure 1 antioxidants-11-00890-f001:**
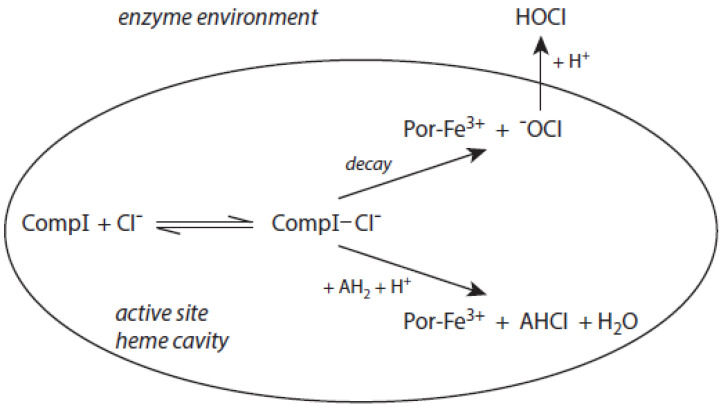
The role of the chloride–Compound I complex (CompI-Cl^−^) in chlorination reactions mediated by MPO according to [44,45]. Explanations are given in the text. CompI denotes Compound I. AH_2_ represents a substrate that binds near the heme pocket. AHCl is the resulting chlorinated substrate.

**Figure 2 antioxidants-11-00890-f002:**
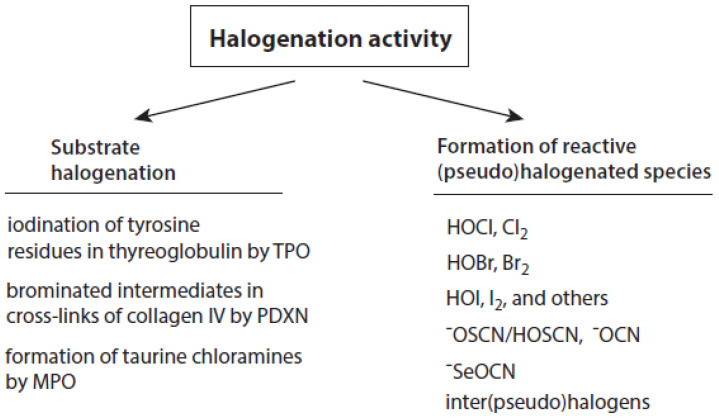
Overview about the halogenation activity of heme peroxidases. Explanations are given in the text.

**Figure 3 antioxidants-11-00890-f003:**
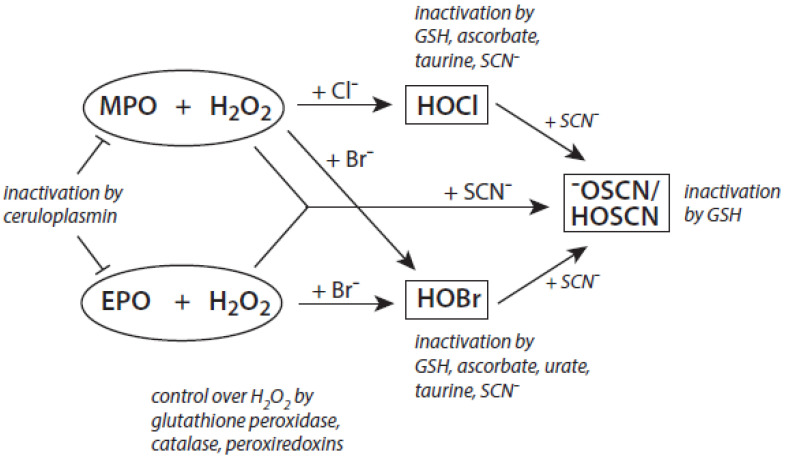
Major antagonizing principles controlling the halogenation activity of MPO and EPO at inflammatory sites. Explanations are given in the text.

**Table 1 antioxidants-11-00890-t001:** Formation of physiologically relevant heme peroxidase-mediated brominated products.

Target Molecule	Brominated Product	Remarks	References
Taurine 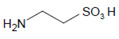	Taurine bromamine 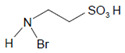	Antimicrobial and anti-inflammatory activity	[183]
Tyrosine (protein-bound) 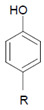	3-Bromotyrosine, 3,5-dibromotyrosine 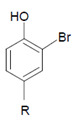 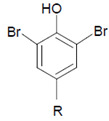	EPO	[122,184]
Uracil 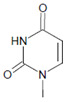	5-Bromouracil 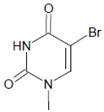	MPO	[185]
2′-Deoxycytidine 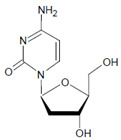	5-Bromo-2′-deoxycytidine 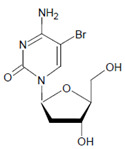	EPO/MPO	[186,187]
2′-Deoxyguanosine 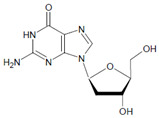	8-Bromo-2′-deoxyguanosine 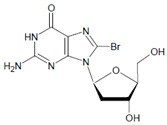	MPO	[188]
Plasmalogens (double bond adjacent to ether linkage) 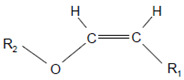	α-Bromo fatty aldehydes 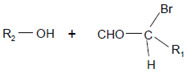	MPO/EPO; other products: lysophospholipids	[144]

**Table 2 antioxidants-11-00890-t002:** Formation of MPO-mediated chlorinated products.

Target Molecule	Chlorinated Product	Remarks	References
Taurine 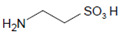	Taurine chloramine 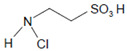	Antimicrobial and anti-inflammatory activity	[183]
Tyrosine (protein-bound) 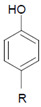	3-Chlorotyrosine 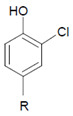		[84,195]
2′-Deoxycytidine 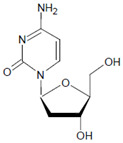	5-Chloro-2′-deoxycytidine 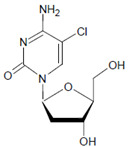		[85,187]
Cytosine 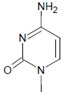	5-Chlorocytosine 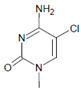		[85]
Uracil 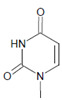	5-Chlorouracil 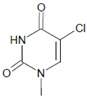		[191,196]
2′-Deoxyguanosine 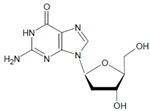	8-Chloro-2′-deoxyguanosine 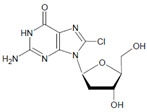		[188]
Plasmalogens(double bond adjacent to ether linkage) 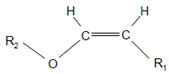	α-Chloro fatty aldehydes 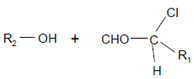	Other products: lysophospholipids	[160,161]
Double bonds in unsaturated lecithins 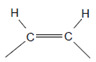	Chlorohydrins at these double bonds 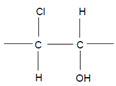 or 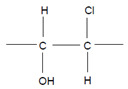		[56,197]
Cholesterol 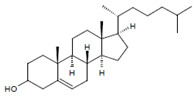	Chlorinated sterols such as 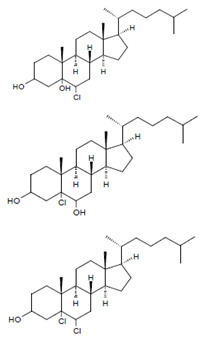	Observed in LDL	[86]

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
