# Peer review of "Halogenation Activity of Mammalian Heme Peroxidases"

_antioxidants, 2022, doi:10.3390/antiox11050890_

Round 1

Reviewer 1 Report

Heme Peroxidases are ubiquitous with diverse biological functions, and as such the subject of this review is of interest. In this review, the authors have described the halogenation activity of this fascinating superfamily of enzymes. Overall, while the review is well researched and the authors have demonstrated an in-depth knowledge of heme peroxidases,  a lack of connection between each session makes it difficult for the readers to understand. This could be improved with more explanation and background  knowledge on heme peroxidase e.g. a description of the overall structure of the enzyme in line 58 (or even a simple schematic) would have prepared the reader better for the rest of the section. Also, basic introduction on Compound I, Compound I and Compound II, in addition to the reactions of each compound would also help.

Author Response

Response to reviewer 1

Thank you very much for your review and helpful comments to improve our manuscript. The following changes as suggested have now been included in the amended version.

=======================

Heme Peroxidases are ubiquitous with diverse biological functions, and as such the subject of this review is of interest. In this review, the authors have described the halogenation activity of this fascinating superfamily of enzymes. Overall, while the review is well researched and the authors have demonstrated an in-depth knowledge of heme peroxidases,  a lack of connection between each session makes it difficult for the readers to understand. This could be improved with more explanation and background  knowledge on heme peroxidase e.g. a description of the overall structure of the enzyme in line 58 (or even a simple schematic) would have prepared the reader better for the rest of the section. Also, basic introduction on Compound I, Compound I and Compound II, in addition to the reactions of each compound would also help.

Response:

1) Subchapter 2.1. has been rewritten, to better express the peculiarities of the heme group in mammalian heme peroxidases.

2) In subchapter 2.2., the major catalytic cycles of heme peroxidases are shortly explained. The title of this subchapter has been extended by major catalytic cycles. Compounds I and II are now better addressed as suggested.

3) Concerning reactions of the peroxidase cycle, we did not go into detail, as the topic of this review is basically focused on halogenation activity.

Reviewer 2 Report

This is a timely and well-written review of the halogenation properties of heme peroxidases. I recommend acceptance after minor revision as noted below:

-Please include in a table the EC codes of the enzymes included in the review.

-In Tables 1 and 2 please include the structural formulae of target and halogenated compounds. 

-In equations 8-11, please replace the double arrow ↔ for the correct equilibrium arrow .

Author Response

Response to reviewer 2

Thank you very much for your review and helpful comments to improve our manuscript. The following changes as suggested have now been included in the amended version of our manuscript.

=================

1) This is a timely and well-written review of the halogenation properties of heme peroxidases. I recommend acceptance after minor revision as noted below:

Response:

Thank you for your comments on our manuscript.

==============

2) -Please include in a table the EC codes of the enzymes included in the review.

Response:

We hesitate to apply EC numbers for the heme peroxidases described in the review for proper reasons. All these enzymes belong to oxidoreductases (EC numbers 1.11.1.x). This superfamily includes also non-heme enzymes. Under 1.11.1.7 (named peroxidases) are listed numerous plant and animal peroxidases including myeloperoxidase, eosinophil peroxidase, and lactoperoxidase. Thus, no differentiation by EC numbers is given for these immunologically important peroxidases. Thyroid peroxidase is mentioned under 1.11.1.8. For peroxidasin no EC number was found. As EC numbering badly reflects the individual properties of mammalian heme peroxidases we did not apply these numbers.

===========================

3) -In Tables 1 and 2 please include the structural formulae of target and halogenated compounds.

Response:

In tables 1 and 2, chemical formula and structural details are now included as suggested.

====================

4) -In equations 8-11, please replace the double arrow ↔ for the correct equilibrium arrow .

Response:

In equations 8 – 11, equilibrium arrows are applied now.

Round 2

Reviewer 1 Report

The authors have done further work on the manuscript which makes the review more enjoyable to read.